# Deposition of Thin Alumina Films Containing 3D Ordered Network of Nanopores on Porous Substrates

**DOI:** 10.3390/ma13132883

**Published:** 2020-06-27

**Authors:** Marija Tkalčević, Marijan Gotić, Lovro Basioli, Martina Lihter, Goran Dražić, Sigrid Bernstorff, Tomislav Vuletić, Maja Mičetić

**Affiliations:** 1Ruđer Bošković Institute, Division of Materials Physics, Bijenička cesta 54, 10000 Zagreb, Croatia; marija.tkalcevic@irb.hr (M.T.); marijan.gotic@irb.hr (M.G.); lovro.basioli@irb.hr (L.B.); 2Laboratory of Nanoscale Biology, Institute of Bioengineering, School of Engineering, EPFL, 1015 Lausanne, Switzerland; martina.lihter@epfl.ch; 3National Institute of Chemistry, Hajdrihova 19, 1001 Ljubljana, Slovenia; goran.drazic@ki.si; 4Elettra-Sincrotrone Trieste, 34149 Basovizza, Italy; sigrid.bernstorff@elettra.eu; 5Institute of Physics, Bijenička Cesta 46, 10000 Zagreb, Croatia; tvuletic@ifs.hr

**Keywords:** nanoporous, microporous, alumina, 3D network of nanopores, membrane

## Abstract

Self-supporting thin films containing nanopores are very promising materials for use for multiple applications, especially in nanofiltration. Here, we present a method for the production of nanomembranes containing a 3D ordered network of nanopores in an alumina matrix, with a diameter of about 1 nm and a body centered tetragonal structure of the network nodes. The material is produced by the magnetron sputtering deposition of a 3D ordered network of Ge nanowires in an alumina matrix, followed by a specific annealing process resulting in the evaporation of Ge. We demonstrate that the films can be easily grown on commercially available alumina substrates containing larger pores with diameters between 20 and 400 nm. We have determined the minimal film thickness needed to entirely cover the larger pores. We believe that these films have the potential for applications in the fields of filtration, separation and sensing.

## 1. Introduction

Nanoporous thin films are widely used for various applications such as energy conversion and storage [1,2,3,4], the selective separation of molecules [5,6,7] and filtration [8,9,10]. Commercially available nanoporous membranes generally exhibit broad size distributions and relatively large thickness. As a result, the materials usually possess poor size-cutoff properties and low transport rates. A separation of similarly sized materials with current commercially available nanoporous membranes is not feasible. Depending on the desired application, the fabrication of membranes with a very small size can be desirable. For instance, in hydrogen separation, the maximum pore size is between 0.3 nm and 0.4 nm [4,11], while for CO_2_ separation, the pore sizes range from 0.3 nm to 1.3 nm [12]. In water desalination, the pore size has a key role in governing the movement of water through the membrane. The diameter of the water molecular cluster is between 0.5 and 0.65 nm while the pore diameters of hydrated Na^+^ and Cl^−^ ions are 0.64 and 0.77 nm, respectively [13,14]. Another very interesting application of ultrathin nanoporous membranes is the conversion of the Gibbs free energy stored in salinity gradients to electricity [15]. Yan, Fei, et al. fabricated an ultrathin silica membrane with a pore diameter in a range between 2 and 3 nm for osmotic energy harnessing [16].

Membrane technology has become a promising industrial alternative compared with traditional treatment techniques, such as distillation, absorption, adsorption, extraction, activated sludge and trickling filters. The separation mechanism is principally based on the size exclusion of matter, i.e., substances with sizes larger than the pores of the membranes are rejected. In order to achieve a high selectivity, the pores in a membrane need to be smaller than the particles in the mixture that we want to separate. The chemical and thermal stabilities are very important to consider when selecting the materials for membranes. For example, polymeric membranes cannot withstand many organic solvents or high-temperature conditions. In these hard conditions, inorganic membranes are usually more suitable. Alumina is a very good material for porous membranes because it is chemically stable, semitransparent, insoluble, inert, and biocompatible [17,18,19]. One of the very attractive applications of nanoporous alumina membranes is their incorporation into a wide variety of sensors [20,21,22,23], for filtration purposes [8,17,24,25] and water treatment [26,27,28,29,30,31].

Many efforts have been made to find a method that would allow the synthesis of uniform, self-supporting alumina membranes with pores in the nano-range. A detailed overview of the preparation of anodic aluminum oxide was given in [32]. During this process, a self-organized, highly ordered array of cylindrically shaped pores is usually produced with a controllable pore diameter. These films are usually prepared by the anodization of aluminum in an appropriate acidic or alkaline solution. With this method, it is possible to fabricate pores whose diameter is as small as 12 nm. The preparation of porous material using organic surfactant molecules is discussed in more detail in [33]. Liu, Yang and coworkers prepared highly permeable alumina membranes by the UV curable technique. The network is formed by the cross-linking of photo curable resins that resulted in rapid solidification of the wet membrane. The most frequent pore size for membranes prepared with this technique was 65.2 nm [34].

Germanium oxidation has been widely investigated in the field of surface science. GeO evaporation is promoted by increasing temperature, since a higher temperature results in a higher vacancy flux toward the surface [35]. The interface reaction is as follows: GeO_2_ + Ge→2GeO↑ [36,37,38]. In our previous work, we have shown that the annealing of regularly ordered inter-connected Ge quantum dot (QD) lattices can result in the formation of regularly ordered void lattices in an Al_2_O_3_ matrix by Ge oxidation [21,22]. The complete desorption of Ge was confirmed using Rutherford backscattering spectrometry measurements. The structural properties of the pores including their size and arrangement, were investigated using grazing incidence small angle x-ray scattering (GISAXS). These measurements confirmed that the structure after annealing is the same as it was before it [39]. In other work, the annealing conditions leading to the formation of a void lattice have been investigated [40]. It was found that complete Ge desorption takes place for a critical annealing air atmosphere pressure of 10^−3^ Pa. For high vacuum annealing conditions, there are not enough oxidants present in the annealing atmosphere for the formation of a GeO_2_ layer, so the Ge loss process from the film is inhibited. The oxidation of Ge QDs in an alumina matrix was already investigated [41]. It was found that Ge QDs in an Al_2_O_3_ matrix are oxidized in large amounts because they are very sensitive to the presence of any oxygen in the deposition chamber.

In this paper, we present a simple method for the fabrication of nanoporous alumina membranes, with a regular arrangement of the pores in a 3D network, having a small pore diameter (~1 nm), and thicknesses of up to 100 nm. The membranes are grown by the magnetron sputtering deposition of thin Ge + Al_2_O_3_ film on commercially available porous alumina membrane with large pore sizes (20–400 nm), followed by specific annealing. In the produced films, the nanopores are arranged in a 3D-ordered continuous network with the arrangement of the nodes controllable by the deposition conditions. The mechanism of thin-film growth over the larger substrate pores is investigated, and a method of reducing the large pore size to the desired value is demonstrated. The presented nanomembranes are interesting for applications in filtering, separation and sensing. 

## 2. Materials and Methods

### 2.1. Preparation Parameters

The materials were produced by co-deposition from Ge (99.999%) and Al_2_O_3_ (99.999%) targets using the magnetron sputtering KJLC CMS-18 system. The thin Ge+Al_2_O_3_ films were simultaneously deposited on three different substrates: quartz, Si (100) substrates, and porous Anodic aluminum oxide (AAO) membranes (Anopore), produced by Whatman (Whatman^®^ Anodisc Inorganic Membranes, provided by Sigma-Aldrich Chemie GmbH, Germany, diameter 47 mm). In the latter case, the films were deposited either on the side of the membrane with smaller pores (20–40 nm) or on the other side having larger pores (about 200 nm). The aim was to cover the AAO membrane with a thin nanoporous alumina film, as illustrated in Figure 1a. Additionally, one film was deposited over the opening of the initial diameter of ~220 nm, in a 20 nm thin SiN_x_ membrane. The membrane was etched in silicon substrate by photolithography, followed by dry and wet etching. The opening was then created by focused ion-beam drilling (FIB). The process was previously published in Feng et al. [42]. The films were prepared on different substrates to allow the use of different characterization techniques.

The substrates were cleaned with acetone, isopropyl alcohol and deionized water in an ultrasound bath before loading into the deposition chamber.

For the deposition of Ge and Al_2_O_3_, we used Direct Current (DC) and Radio Frequency (RF) magnetrons, respectively. The Ge sputtering power was tuned from 3 W to 13 W, while the power of the Al_2_O_3_ sputtering was kept constant at 140 W. The argon pressure was 3.5 Torr for all the films, and the deposition temperature was 300 °C. The deposition parameters for all the films are given in Table 1. Two series of films were deposited, differing by the Ge sputtering power (denoted by letter P) and by the deposition time (denoted by letter D). Thus, the name of each film consists of two letters denoting the deposition time and Ge power (for example, P1D1), as shown in Table 1. Additional films having the same sputtering powers and longer deposition times were prepared for some of the measurements to ensure better statistics.

After the deposition, the films consisted of a 3D mesh of Ge nanowires embedded in an Al_2_O_3_ matrix. In order to achieve Ge evaporation, a heat treatment was performed in a low vacuum (10^−2^ mbar) environment in an electric furnace for lab heat treatment. After the thermal annealing for 4 min at 630 °C, the Ge was removed from the sample leaving the porous structure behind. The color of the films on quartz substrate and AAO membrane changed during the annealing from brown, which is caused by the Ge presence, to transparent after the Ge evaporation, as visible from Figure 1b. In order to be able to clearly distinguish the films and substrates, we use the following notations: (i) Ge-based film for the as-deposited, non-porous film; (ii) nanoporous film for the annealed Ge-based film, which has nanopores that we use to cover the substrate pores; and (iii) AAO substrate membrane for the anodic alumina with the large pores.

### 2.2. Characterization Techniques

The grazing incidence small-angle x-ray scattering (GISAXS) measurements were performed at the synchrotron Elettra, Trieste, Italy at the SAXS beamline using a photon energy of 8 keV and a Pilatus3 1M detector. The films deposited on non-porous Si substrates were used only for the GISAXS measurements. We expect the same structure of the films grown on membranes and SiN_x_ membrane because they grow with the same mechanism independently on the substrate. The GISAXS maps were numerically analyzed using the procedure described in [43,44,45]. 

The transmission electron microscopy (TEM) of the cross-section of the film was performed using a JEOL ARM 200 CF scanning transmission electron microscope (STEM), operated at 200 kV and equipped with a field-emission gun and a high-angle annular dark-field detector (HAADF) for Z-contrast imaging.The TEM of the opening in the SiN_x_ membrane overgrown by the film was performed using Talos TEM (ThermoFisher Scientific, Hillsboro, OR, USA) at 200 kV.

The scanning electron microscopy (SEM) was performed using the thermal field emission scanning electron microscope (FE SEM, model JSM-7000 F) manufactured by JEOL Ltd (Tokyo, Japan). FE SEM was linked to the EDS/INCA 350 (energy-dispersive X-ray analyzer) manufactured by Oxford Instruments Ltd. (Abingdon, UK).

## 3. Results

### 3.1. Structural Properties of Thin Nanoporous Films

In this section, we analyze the nanostructure of the films itself. In the next section, we investigate how they cover the substrate having larger pores.

The typical structural properties of the prepared films are demonstrated in Figure 2. The model of an ideal network structure is shown in Figure 2a, while the main parameters of it are illustrated in Figure 2b. The nanopores make a 3D network having a body centered (BCT) tetragonal arrangement of the nodes. The separation between the nodes in the plane parallel to the substrate is described by the parameter *a*, while the vertical separation is *c*. The pores have the radius *R* and the length *L*. The TEM images of the film structure are shown in Figure 2c,d. The shown films differ by the Ge sputtering power, resulting in the formation of longer, more separated pores for the smaller Ge power. The networks are continuous, which is ensured by the preparation method, i.e., the evaporation of Ge from the film during the sample annealing. The insertion of even a very thin (1 nm) layer of pure matrix prevents Ge oxide evaporation, because the network is not continuous in that case. Therefore, the same preparation method is not efficient for Ge quantum dot multilayer films. In such films, Ge quantum dots are fully surrounded by alumina matrix, which prevents the evaporation of Ge. Therefore, these films cannot be used for the creation of nanoporous films. The GISAXS intensity patterns of the films’ structure are shown in Figure 2e,f. The two characteristic lateral peaks are related to the arrangement of the network nodes and to their regularity. A more regular network produces stronger and narrower Bragg peaks, so the P1D3 film has a better ordering in this case. More details about the GISAXS maps of such structures are given in [43,44,45]. GISAXS is very suitable for the analysis of these structures, as it provides data about nanopore ordering and size properties with excellent statistics.

The GISAXS maps of all the investigated films before and after annealing (i.e., with Ge quantum wires and nanopores, respectively) are given in Figure 3. The first three films (Figure 3a–c) differ only by the film thickness (D1–D3), while the last two (Figure 3d,e) have different Ge sputtering powers (P2, P3), i.e., different pore arrangements and size properties. The side maxima in the GISAXS maps (Bragg spots) are at nearly the same positions for the first three films, showing the homogeneity of the structure, because these films differ only by their thickness. For the last two films, the spots become more elongated and more spaced, showing an increase in the disorder and a decrease in the lattice parameters of the formed 3D pore network.

All the films except the last one have nearly the same position of the Bragg spots before and after annealing (first and second rows in Figure 3, respectively), showing that during annealing, the Ge left the film, leaving empty space behind without affecting the alumina. The GISAXS method is sensitive to the electron density contrast between the nano-objects and the matrix. The Ge and alumina have a strong difference in electron density, as well as in the empty space (pores) and alumina. Therefore, we see nearly the same GISAXS maps from the Ge nanowires as from the pores of the same shape in the alumina matrix. Only the last film changed the structure during annealing, probably due to the high percentage of Ge that caused a collapse of the structure after leaving the film. This follows clearly from the GISAXS maps of that film (P3D1, shown in Figure 3c). The GISAXS map of the annealed film (P3D1 annealed) differs significantly from the map of the as-grown film. This fact shows clear evidence that the internal structure changed significantly during annealing. The lateral peaks called Bragg spots are clearly visible in the As-grown film, while they are absent for the annealed one, showing that there is no regular ordering of the pores after annealing.

The GISAXS maps were numerically analyzed using the procedure described in [44,45] in order to obtain the values of the structural parameters of the pores and their arrangement. The results of the analysis are given in Table 2. From the results, we can see that the nanopore radius (*R*) is around 0.6 nm for all the films. This is due to the same deposition temperature for all the films, as shown in the case of Ge quantum wires in [46]. The deposition time (D1–D3) only significantly influences the thickness of the films (D), assuming a constant Ge sputtering power (P1), while the pore length (*L*) depends on the Ge sputtering power during the film preparation (P1–P3). The pore length decreases with increasing Ge sputtering power. We want to point out here that the pore length *L* refers to the length of the pore between the two adjacent nodes of the BCT lattice (please see Figure 2b), but the pores are interconnected, so the real pore length is much larger. The pore length L is not correlated to the film thickness. The film grows homogeneously with the same value of the pore length. More information about the production of Ge quantum wire lattices with different structural parameters is given in [46].

In summary, the structural properties of the nanopores are determined by the properties of the Ge quantum wire lattices. By tuning the deposition parameters, it is possible to produce networks of nanopores having different radii and lengths.

### 3.2. Growth of Nanoporous Thin Films on Alumina Substrate with Larger Pores

In this section, we investigate the preparation of nanomembranes using the above-mentioned films. As described in the Methods section, we deposited the films on porous alumina substrates (Whatman^®^ Anodisc Inorganic Membranes) having larger pores. The used substrate membranes have different pore sizes on their sides as visible from the SEM measurements, shown in Figure 4. Side 1 has pores with a diameter of 20–40 nm (Figure 4a), and Side 2 has larger pores with a diameter of about 200 nm (Figure 4b).

The effect of the film deposition on Side 1 of the membrane is shown in Figure 5. The SEM images of the film’s surface clearly show the coverage of the pores with increasing film thickness *D*. The thinnest film P1D1 shown in Figure 5a does not cover the pores. The coverage is complete for the middle thickness (40 nm) shown in Figure 5b. This indicates that the minimal film thickness needed to achieve the full coverage of the substrate holes is found to be nearly equal to the lateral size of the holes. However, features similar to cracks are visible in the film structure. A complete coverage and a smooth film surface are achieved for the film P1D3 having the largest thickness of about 80 nm.

The coverage of the larger pores (Side 2) of the AAO substrate by the films is shown in Figure 6. The films differing by the deposition time are shown in Figure 6a,b, and the films differing by Ge sputtering power are shown in Figure 6c,d. The pores are not fully covered in all cases; however, the pore sizes are strongly reduced for the longest deposition time (P1D3) and for the largest Ge sputtering power (P3D1). Since the reduction of the pore size is proportional to the film thickness, this technique can be applied for tailoring the pore size to desired value. 

The effect of the pore coverage is also investigated on a single opening (pore) in the SiN_x_ membrane; see Figure 7. The single opening of initial diameter of ~220 nm was investigated by TEM to inspect the overgrowth behavior of the films on different substrates. Figure 7a shows the opening having a diameter of ~120 nm after the growth of the film P3D1 with the thickness of 42 nm. The non-covered part of the opening is visible as a circular shape in light-gray color. The grown film appears slightly darker, while the rest of the membrane exhibits a dark gray color. The effect is better visible in Figure 7b where the enlarged part of the opening is shown. The nanopores in the film are visible, and the thickness of the covered part of the opening is about 40 nm. As visible from the images, the film narrowed the opening for the value of its thickness, shrinking the pore down to ~120 nm. The nanoporous structure of the film is shown in Figure 7c. Results for the reduction of the pore size in the SiNx membrane shown in Figure 7a–c, are shown in Figure 7d. The curve shows a negative slope with the value of 2.26, which may be used for the estimation of the pore size reduction using the presented technique.

## 4. Conclusions

We have demonstrated a simple method to produce self-supporting alumina nanomembranes with a 3D-ordered network of nanopores having a body-centered tetragonal ordering of its nodes and a pore diameter of about 1 nm. The membranes are produced by the magnetron sputtering deposition of thin films consisting of a 3D network of Ge quantum wires in an alumina matrix. The films are deposited on a large-pore alumina substrate, followed by a specific annealing process. The annealing induces Ge evaporation, resulting in the formation of a 3D network of nanopores. The film covers the larger pores, and the minimal film thickness needed to achieve the full coverage of the substrate holes is nearly equal to half of the lateral hole size. However, thicker films ensure defect-free pore coverages. This method can also be used for reducing larger pore sizes to a desired value. The pore sizes and inter-pore distance can be controlled by the deposition conditions. We believe that these nanomembranes could be used for sensing applications and filtration and separation purposes. Additionally, the same material can be produced using other volatile elements instead of Ge, and that will be the topic of our future work.

## Figures and Tables

**Figure 1 materials-13-02883-f001:**
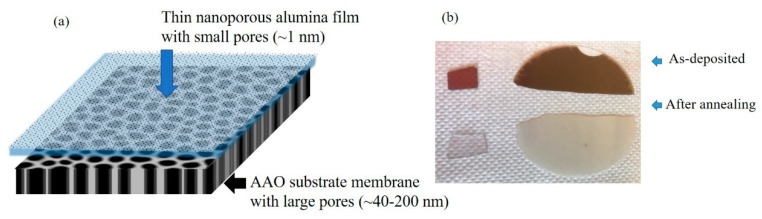
(**a**) Scheme of the material structure. Thin nanoporous alumina film (pore size ~1 nm) is deposited on the porous anodic aluminum oxide (AAO) substrate with large pores (40–200 nm). (**b**) Image of the prepared thin films deposited on porous AAO (right) and quartz (left) before and after annealing. The change in color is due to the evaporation of Ge out of the film during annealing.

**Figure 2 materials-13-02883-f002:**
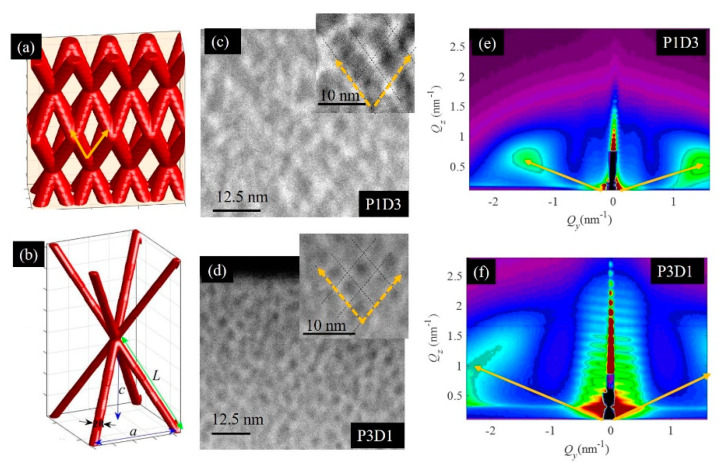
(**a**) 3D model of the nanoporous network. (**b**) Scheme of the main structural parameters of this network. (**c**,**d**) STEM-HAADF micrographs of the P1D3 and P3D1 films’ cross-sections, respectively, and (**e**,**f**) their grazing incidence small angle x-ray scattering (GISAXS) maps. The yellow dashed arrows indicate the direction of the nanopore ordering. The full yellow arrows indicate the positions of Bragg spots in the GISAXS maps that are related to the ordering of the nanopores in 3D networks.

**Figure 3 materials-13-02883-f003:**
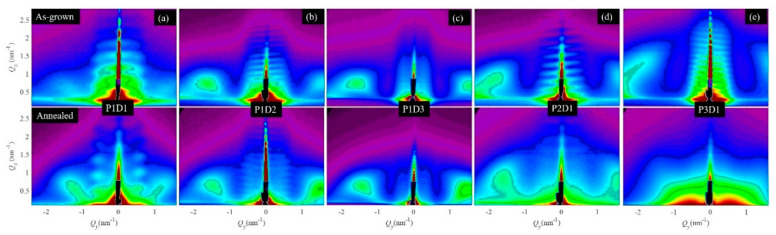
GISAXS maps of the investigated films after deposition (upper row) and after Ge evaporation by suitable annealing. (**a**)Film P1D1, (**b**) film P1D2,(**c**) film P1D3,(**d**) film P2D1,(**e**) film P3D1, please see Table 1 for deposition details of each film.

**Figure 4 materials-13-02883-f004:**
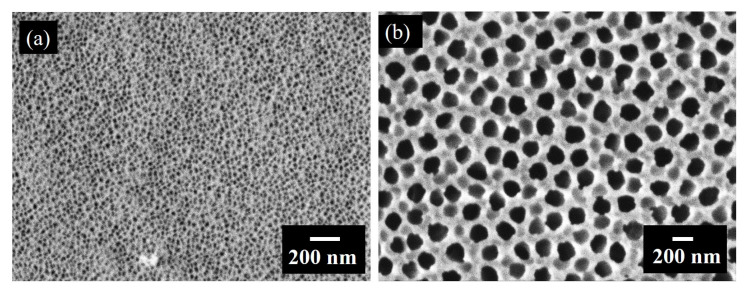
SEM micrographs of the substrate used for the deposition of thin films. (**a**) Side 1 with smaller pores of size of ~40 nm. (**b**) Side 2 with larger pores of size ~200 nm.

**Figure 5 materials-13-02883-f005:**
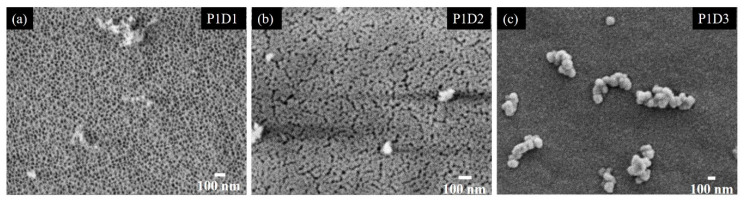
SEM micrographs of the films deposited on Side 1 of the substrate. The films differ by their thickness (D1 (**a**), D2 (**b**), D3 (**c**)) as the deposition time increases from left to right.

**Figure 6 materials-13-02883-f006:**
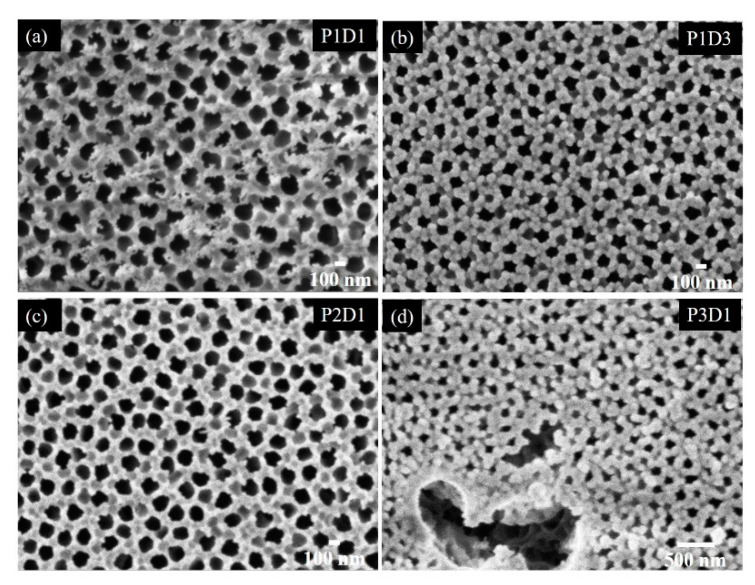
SEM micrographs of films grown under different conditions on Side 2 of the porous AAO substrate. The partial covering of the pores is well visible. (**a**) Film P1D1, (**b**) film P1D3, (**c**) film P2D1, (**d**) film P3D1, for the deposition details of each film please see Table 1. The large pores of the AAO substrate are partially visible within the defect in the thin film structure visible in panel (**d**)

**Figure 7 materials-13-02883-f007:**
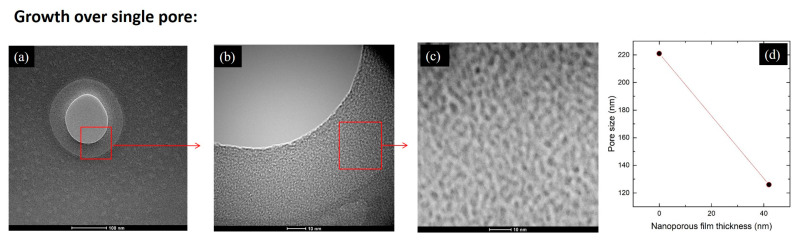
(**a–c**) TEM micrographs of the film grown over a single opening in 20 nm thin SiN_x_ membrane taken at different magnifications. The initial diameter of the opening prior to growth was approximately 220 nm. The enlarged areas are denoted by red lines. (**d**) Dependence of the opening size on the thickness of deposited nanoporous film.

**Table 1 materials-13-02883-t001:** Deposition parameters of the films; *t*_D_ denotes the deposition time, *P*(Ge) denotes the Ge sputtering power.

Name/Par	$t_{D}/\min$	*P*(Ge)/W
P1D1	30	3
P1D2	60	3
P1D3	120	3
P2D1	30	6
P3D1	30	13

**Table 2 materials-13-02883-t002:** Structural parameters of the investigated nanopore membranes: *a,c* are in plane and vertical separation of the nanopore nodes, respectively. *R* is the nanopore radius, *L* is their length (separation between two nearest nodes), and *D* is the film thickness. All values are given in nm.

Name/Par	*a*	*c*	*R*	*L*	*D*
P1D1	2.9	4.2	0.6	4.7	20
P1D2	3.1	4.0	0.6	4.7	40
P1D3	3.4	4.1	0.6	4.7	80
P2D1	3.0	3.3	0.6	3.9	26
P3D1	2.8	2.4	0.6	3.1	42

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
