# Peer review of "Deposition of Thin Alumina Films Containing 3D Ordered Network of Nanopores on Porous Substrates"

_materials, 2020, doi:10.3390/ma13132883_

Round 1

Reviewer 1 Report

This manuscript deals with the formation of nanoporous films deposited on different supports by using Ge sputtering. It is interesting and well written and so it deserves publications in Materials after addressing the following comments:

- Line 26: replace “conversation” by “conversion”

- Lines 101-102: Please, give additional experimental details. Were Ge and Al2O3 sputtered together? Please, give conditions for Al2O3 sputtering.

- Were GISAXS maps carried out in films deposited on porous alumina and SiNx membrane to confirm its structure?

- Have the authors some proofs concerning the closure of pores in SiNx membranes by increasing deposition time?

Author Response

1.) 2Lines 101-102: Please, give additional experimental details. Were Ge and Al2O3 sputtered together? Please, give conditions for Al2O3 sputtering.

Re: We have added the details of the deposition (line 110):

„For the deposition of Ge and Al2O3 we have used DC and RF magnetrons respectively. The Ge sputtering power was tuned in the range from 3 W to 13 W, while the power of Al2O3 sputtering was kept constant at 140 W.“

2.) Were GISAXS maps carried out in films deposited on porous alumina and SiNx membrane to confirm its structure?

Re: For the GISAXS measurements we have to use film deposited on flat substrate (like Si wafer or quartz). Therefore, the measurements were not possible on the alumina membranes due their curvature.  The measurement on SiNx membrane was impossible due to the very small size of the pore. The probing beam for GISAXS should be smaller than the pore size to probe only the area above the hole, what was not possible. However, our previous investigations on these films showed that the structure of films is the same regardless the substrate roughness. They grow with the same mechanism independently on the substrate. In addition, the TEM measurements confirmed that the structure is very similar on the membrane and on the flat substrate.

We have added a sentence in Experimental to pint out these findings (line 137):„The films deposited on non-porous Si substrates were used only for the GISAXS measurements. We expect the same structure of the films grown on membranes and SiNx membrane, because grow with the same mechanism independently on the substrate.“

3.) Have the authors some proofs concerning the closure of pores in SiNx membranes by increasing deposition time?

Re: The focus of this work is covering the larger pores in alumina membranes. Therefore, we have deposited only one film on the SiNx membrane, just to have an additional source of evidences for the overgrowth of the film of the pores on different substrates. However, the pores in alumina membrane are fully covered. We have added the following text to the paper to clarify these findings (line 251): „Only a single pore was investigated to prove the same overgrowth behavior of the films on different substrates.“

Reviewer 2 Report

The author described the magnetron sputtering deposition preparation of the self-supporting 3D ordered porous alumina membrane templated by sacrificed Ge. The pore structure topology created is interesting. However, there are many issues impede the acceptance of the manuscript.

  • It is suggested to distinguish the usage of membrane and film, the former denotes the porous one, the latter denotes nonporous one, to my best knowledge. However, the authors mix the usage of them.
  • I’m fail to find the delamination experiment to demonstrate the author’s “self-support” claim.
  • The purposes of experiment design are not clear. The attempts to deposit the porous alumina on the nonporous substrate is useless as membrane. The deposition on the porous alumina or porous SiNx is rather a tool to modify the substrate pore size than to prepare a new porous membrane.
  • Either Ge or magnetron sputtering deposition are pretty expensive. If the authors fail to present a unique property or highly value-added application of the membrane, the process is uneconomical.
  • Why the authors not choose cheaper metals such as Zn? The zinc or zinc oxide are also volatile under the high temperature that render it as sacrificed agent to making pores.

Author Response

(1) It is suggested to distinguish the usage of membrane and film, the former denotes the porous one, the latter denotes nonporous one, to my best knowledge. However, the authors mix the usage of them.

Re: We agree with the Referee. The as-deposited film is not porous; the film became porous after thermal annealing. We use the alumina membrane with large pores as substrate, so we actually need to distinguish these three items. Therefore, we use the following terms: (i) Ge-based film for the as-deposited, non-porous film, (ii) nanoporous alumina film for the annealed Ge-based film which has nanopores which we use to cover the substrate pores, and (iii) AAO substrate membrane for the anodic alumina with the large pores. We have added the following text to pint out these items (line 116): „ In order to be able to clearly distinguish the films and substrates we use the following notations: (i) Ge-based film for the as-deposited, non-porous film, (ii) nanoporous film for the annealed Ge-based film which has nanopores which we use to cover the substrate pores, and (iii) AAO substrate membrane for the anodic alumina with the large pores.“

(2) I’m fail to find the delamination experiment to demonstrate the author’s “self-support” claim.

Re: Delamination simulation experiments were not performed for the investigated thin films. Self-supporting is based on the fact that it covers large pores without support. We have changed the title of the paper to avoid further confusions. The new title is “Deposition of thin alumina films containing 3D ordered network of nanopores on porous substrates

(3) The purposes of experiment design are not clear. The attempts to deposit the porous alumina on the nonporous substrate is useless as membrane. The deposition on the porous alumina or porous SiNx is rather a tool to modify the substrate pore size than to prepare a new porous membrane.

Re: The films were deposited on the non-porous substrate only because we need them for the GISAXS measurements to determine the structure of the films. These measurements are not possible on the porous membranes because they are not flat enough for the measurements.

The purpose of the experiment is to produce membrane with the nanopore openings, useful for different applications. We have added the following text to clarify the usage of different substrates (line 137): „ The films deposited on non-porous Si substrates were used only for the GISAXS measurements. We expect the same structure of the films grown on membranes and SiNx membrane, because grow with the same mechanism independently on the substrate. “

(4) Either Ge or magnetron sputtering deposition are pretty expensive. If the authors fail to present a unique property or highly value-added application of the membrane, the process is uneconomical.

Re: We do not fully agree with the referee. First we have demonstrated the production of thin film membrane. Due to its nanometer sized thickness, the amount of Ge needed to produce large area of this membrane is very small. Second, the magnetron sputtering is widely used in fabrication of thin-film coatings, i.e. for smart window production, so we do not believe that it is fully uneconomical. Thirdly, the purpose of this work is fully scientific. We believe that the formation of 3D ordered mesh of nanopores helps development of high-value materials.

(5) Why the authors not choose cheaper metals such as Zn? The zinc or zinc oxide are also volatile under the high temperature that render it as sacrificed agent to making pores.

Re: We thank the Referee for the very interesting suggestion, we will try to repeat the experiment in the future with Zn. A sentence indicating our future directions of investigations is added in the conclusions (line 281): “Additionally, the same material can be produced using other volatile elements instead of Ge, and that will be the topic of our future work.”

Reviewer 3 Report

The Authors report on a pore texturing method based on the simultaneous sputtering of GeOx and Alumina on different substrates included a pre patterned Alumina matrix.

The methodology and mechanisms are clearly explained, results are interesting although not particularly surprising.

Here are my comments on how to possibly improve the work:

1) Starting from the title, while I understand the meaning, I have some doubts in the association of the words grown and nanopores, I would suggest to reformulate it because to "grow a pore" is a kind of non-sense. The same comment applies to line 186 where the authors speak about "the structural proprieties of nanopores"!

2) The main application discussed in the text is about filtration, the MS discuss a wide and satisfying review of porous membranes, however, I have some doubts that a 1 nm nanopore matrix can be effectively used for any filtration application; the authors should better evidence the advantage of having a so small pore membranes. I would also have appreciated a filtration performance of the material they propose.

3) In the materials section I don't see the description of the different powder targets (P1-3).

Figure 1 is not effective, there is no indication of sizes, the dotted "Deposited film" in vague, the underlying material substrate is not indicated as a substrate and it may be understood that the vertically aligned pores are the result. This figure should visually guide the reader on the DOE and the goal of the study, and this is not the case here.

4) Line 142 I don't understand what is a "Ge quantum dot multilayer film" nor why the methodology here described would not apply to it.

5) In Fig3 the authors say that the structure collapses in P3 sample, however I do not understand clearly how this is visible in in the GISAXS maps evolution. I would suggest the Authors to better explain how to extract information from the maps to get the interest from a broader readership.

6) Please clarify how the pore length and the membrane thickness are related, I understood that the pore goes through the whole membrane thickness. This is to understand the parameter L in Table 2.

7) For the last section of the MS the authors show the pore reduction/filling on a pre-patterned AlOx substrate. The main message is that the sputtered layer thickness correlates with the pore diameter reduction, so I would add a plot of the membrane thickness vs pore size. And the nanopore diameters for a completely covered substrate.

8) What are the particles fig5c?

Author Response

  1. Starting from the title, while I understand the meaning, I have some doubts in the association of the words grown and nanopores, I would suggest to reformulate it because to "grow a pore" is a kind of non-sense. The same comment applies to line 186 where the authors speak about "the structural propreties of nanopores“

Re: We have changed the title: “Deposition of thin alumina films containing 3D ordered network of nanopores on porous substrates

2. The main application discussed in the text is about filtration, the MS discuss a wide and satisfying review of porous membranes, however, I have some doubts that a 1 nm nanopore matrix can be effectively used for any filtration application; the authors should better evidence the advantage of having a so small pore membranes. I would also have appreciated a filtration performance of the material they propose.

Re: The filtration performance will be the subject of our future work. At the moment we investigate the small defects that are present in the films and how to get rid of them by optimizing the deposition parameters. Therefore, we cannot test accurately the performance of the membranes, and more investigation is needed before the membranes will be the ‘final’ product. However, we have added several references showing that such materials can be used for the filtration. The following text has been added (line 31): “ Depending on the desired application fabrication of membranes with a very small size can be desirable. For instance, in hydrogen separation maximum pore size is between 0.3 nm and 0.4 nm [4,11] while for CO2 separation pore sizes are ranging from 0.3 nm to 1.3 nm [12]. In water desalination, the pore size has a key role in governing the water through the membrane. The diameter of the water molecular cluster is between 0.5 and 0.65 nm while pore diameters of hydrated Na+ and Cl- ions are 0.64 and 0.77 nm respectively [13,14]. Another very interesting application of ultrathin nanoporous membranes is the conversion of the Gibbs free energy stored in salinity gradients to electricity [15]. Yan, Fei, et al. fabricated an ultrathin silica membrane with a pore diameter in a range between 2-3 nm for osmotic energy harnessing [16].”

3.) In the materials section I don't see the description of the different powder targets (P1-3)

Re: We have changed the description of the sample preparation to be more clear. Only two targets were used: Ge and alumina. To obtain different structure of the nanporous alumina films we have changed the sputtering powers of Ge target. The new text (line 110): “For the deposition of Ge and Al2O3 we have used DC and RF magnetrons respectively. The Ge sputtering power was tuned in the range from 3 W to 13 W, while the power of Al2O3 sputtering was kept constant at 140 W”

4.) Figure 1 is not effective, there is no indication of sizes, the dotted "Deposited film" in vague, the underlying material substrate is not indicated as a substrate and it may be understood that the vertically aligned pores are the result. This figure should visually guide the reader on the DOE and the goal of the study, and this is not the case here.

Re: We have changed Fig. 1 as suggested, and we have also changed the figure caption accordingly.

5.) Line 142 I don't understand what is a "Ge quantum dot multilayer film" nor why the methodology here described would not apply to it.

Re: The quantum dot multilayer film differ from the Ge nanowire network because Ge quantum dots are fully surrounded by alumina matrix. Therefore, Ge from quantum dots cannot evaporate during annealing and form porous structure. On the other side, Ge quantum wires are interconnected, and due to the interconnection Ge evaporates from the entire film during annealing. We have added the following text (line 162): “Therefore, the same preparation method is not efficient for Ge quantum dot multilayer films. In such films Ge quantum dots are fully surrounded by alumina matrix which prevent evaporation of Ge. Therefore, these films cannot be used for creation of nanoporous films.”

6.) In Fig3 the authors say that the structure collapses in P3 sample, however I do not understand clearly how this is visible in in the GISAXS maps evolution. I would suggest the Authors to better explain how to extract information from the maps to get the interest from a broader readership.

Re: We have explained it better in the text. The following text has been added (line 192): “This follows clearly from the GISAXS maps of that film (P3D1, shown in Fig. 3 (c)). The GISAXS map of the annealed film (P3D1 Annealed) differs significantly from the map of the As-grown film. This fact shows clear evidence that the internal structure changed significantly during annealing. The lateral peaks called Bragg spots are clearly visible in As-grown film, while they are absent for the annealed one, showing that there is no regular ordering of the pores after annealing.”

7.) Please clarify how the pore length and the membrane thickness are related, I understood that the pore goes through the whole membrane thickness. This is to understand the parameter L in Table 2.

Re: The pore length and membrane thickness are not correlated. The pore length refers to the length of the pore between two adjacent nodes off the 3D mesh, and is usually significantly smaller than the film thickness. The thickness of the nanoporous film is determined by the deposition time, while the pore length L is determined by the sputtering power of Ge and alumina.  We have added the following text to clarify these items (line 209):” We want to point out here that the pore length L refers to the length of the pore between the two adjacent nodes of the BCT lattice (please see Fig. 2(b)), but the pores are interconnected, so the real pore length is much larger. The pore length L is not correlated to the film thickness. The film grows homogeneously with the same value of the pore length”

8.) For the last section of the MS the authors show the pore reduction/filling on a pre-patterned AlOx substrate. The main message is that the sputtered layer thickness correlates with the pore diameter reduction, so I would add a plot of the membrane thickness vs pore size. And the nanopore diameters for a completely covered substrate.

Re: We have added a new panel in Figure 7 (Fig. 7(d)) with the reduction of the pore size vs film thickness. The added text in the Figure caption is: ”Dependence of the membrane pore size on the thickness of the deposited nanoporous film.”

The following texts has been added (line 264): “Finally, we estimate the size-reduction of the substrate pores by the deposition of thin film using the presented microscopy data. Although, we do not have enough data for the reliable characterization of the pore reduction, in Fig. 7(d) are presented the results for the reduction of the pore size of the pore in the SiNx membrane shown in Fig. 7(a)-(c). The curve shows a negative slope with the value of 2.26 which may be used for the estimation of the pore size reduction using the presented technique.”

9.) What are the particles fig5c

Re: We believe that the particles are Ge because only three elements are deposited Al, O and Ge. We know that Ge evaporates from the substrate so i probably form some structures on the surface.

Round 2

Reviewer 3 Report

I can see an improvement in the revised manuscripts which looks to me suitable for publication in its present form.

Regards